# k-Fold Cross-Validation Can Significantly Over-Estimate True Classification Accuracy in Common EEG-Based Passive BCI Experimental Designs: An Empirical Investigation

**DOI:** 10.3390/s23136077

**Published:** 2023-07-01

**Authors:** Jacob White, Sarah D. Power

**Affiliations:** 1Faculty of Engineering and Applied Science, Memorial University of Newfoundland, St. John’s, NL A1B 3X5, Canada; jrw111@mun.ca; 2Faculty of Medicine, Memorial University of Newfoundland, St. John’s, NL A1B 3V6, Canada

**Keywords:** EEG, time-series, cross validation, passive brain–computer interface

## Abstract

In passive BCI studies, a common approach is to collect data from mental states of interest during relatively long trials and divide these trials into shorter “epochs” to serve as individual samples in classification. While it is known that using k-fold cross-validation (CV) in this scenario can result in unreliable estimates of mental state separability (due to autocorrelation in the samples derived from the same trial), k-fold CV is still commonly used and reported in passive BCI studies. What is not known is the extent to which k-fold CV misrepresents true mental state separability. This makes it difficult to interpret the results of studies that use it. Furthermore, if the seriousness of the problem were clearly known, perhaps more researchers would be aware that they should avoid it. In this work, a novel experiment explored how the degree of correlation among samples within a class affects EEG-based mental state classification accuracy estimated by k-fold CV. Results were compared to a ground-truth (GT) accuracy and to “block-wise” CV, an alternative to k-fold which is purported to alleviate the autocorrelation issues. Factors such as the degree of true class separability and the feature set and classifier used were also explored. The results show that, under some conditions, k-fold CV inflated the GT classification accuracy by up to 25%, but block-wise CV underestimated the GT accuracy by as much as 11%. It is our recommendation that the number of samples derived from the same trial should be reduced whenever possible in single-subject analysis, and that both the k-fold and block-wise CV results are reported.

## 1. Introduction

A brain–computer interface (BCI) is a system that translates information obtained via neurophysiological signals into commands for controlling an external device. Active BCI systems aim to provide individuals with severe motor disabilities a movement-free means of communication and environmental control by translating intentionally-modulated patterns of brain activity into control commands for external devices. Passive BCI (pBCI) systems are potentially more broadly applicable; such systems are not for the intentional control of external devices, but rather they aim to enhance human–computer interactions by providing implicit information about the user’s mental state (e.g., cognitive or emotional) [1]. An example would be a system that monitors a driver’s level of fatigue/alertness and provides an alarm when a state of drowsiness is detected. Due to the advantages it offers in terms of non-invasiveness, cost-effectiveness, portability, and high temporal resolution, the most common technique used in BCI research for obtaining the neurophysiological signals is electroencephalography (EEG) [2,3].

Typically, a practical BCI system, whether active or passive, works as follows: (1) signals are acquired from the user via EEG, (2) the raw signals are pre-processed to remove unwanted artifacts, (3) features of the signals that are useful in discriminating the different mental states the BCI is meant to detect are calculated over many short segments/epochs of EEG to create samples, (4) these feature data are fed to a machine learning algorithm trained on previously collected neural data to classify/predict the current mental state, and (5) appropriate commands are sent to modify the connected external device, as appropriate. This process is depicted in Figure 1.

The description of a practical BCI given above describes real-time, or “online”, classification. In online classification, the training data have been previously collected and used to pre-train the classifier, and then the user’s mental state is predicted in real-time as they use the device. The performance of the classifier can be assessed based on the accuracy of those real-time predictions versus the user’s true state. However, when investigating BCI systems for new applications (e.g. when considering previously unexplored mental states, user populations, or environmental conditions) or when testing/comparing new classification algorithms even in established applications, it is often more useful or convenient to perform the classification “offline”. In offline analysis, a relatively large amount of neural data are collected from a group of participants that includes the different mental states of interest and any relevant experimental conditions, and the data are then stored to be analyzed later; there is no “real-time” prediction of the user’s mental state. This allows the researcher to explore and compare different techniques for classifying the mental states of interest, or examine the effects of different experimental conditions, much more efficiently than would be possible with online analysis. Offline analysis is often used in proof-of-concept studies to determine the best approaches to use in subsequent online studies.

### Cross-Validation Techniques for Offline Classification

One important consideration in offline studies is how the data are divided into training and test sets to estimate the classifier performance. This can be performed in a single train/test split of the samples or by cross-validation (CV). The most common CV technique is k-fold CV, in which the full dataset is randomly partitioned into *k* subsets of samples, and one of those subsets is retained for testing the classifier, while the remaining *k* −1 subsets comprise the training set. This process is repeated *k* times until all subsets (and thus, all individual samples) have been used for testing the classifier exactly once. The overall classifier performance is then estimated as the average of the resulting *k* classification accuracies from each step of the CV. Because there can be significant variation in the accuracy obtained with different train/test splits, this method yields a more generalizable estimate of classifier performance than taking just a single split.

While k-fold cross-validation is a very commonly used technique for evaluating machine learning algorithms offline, it can present issues with some types of data when the samples within classes are collected in close proximity in time, without randomization with the other class(es). For time-series data, similar to EEG, the process of randomly dividing all samples into *k* partitions results in the training and test sets containing samples from the same class that are highly correlated due to their proximity in time. This violates the assumption of independence that is critical to the validity of k-fold cross-validation [4]. The result is that the classifier could pick up differences between the classes that are actually just related to this temporal correlation of some samples, rather than to any true class-related difference.

This is typically not an issue in active BCI research, where most often (1) the trials collected are relatively short (less than 10 s) and, thus, only one EEG sample/epoch is calculated from each trial, and (2) the trial order of the mental states of interest (usually different mental tasks, such as motor imagery of different body parts) is randomized. However, for passive BCI studies, where the mental state (e.g., mental workload, fatigue) data often must be collected over longer trials (e.g., 30 s up to several minutes), this is a common issue. In such studies, in order to produce a sufficient number of samples in a reasonable period of time, multiple EEG samples/epochs are calculated from a single trial. For example, for a trial 1 min in duration that represents a single mental state (e.g., high mental workload), it would be common to extract 60 individual samples calculated over consecutive 1-s, non-overlapping epochs. However, due to their proximity in time, these samples will likely be more highly correlated with one another than they would with samples from other trials, regardless of their class membership. As such, in a k-fold cross-validation analysis, when some of these 60 samples end up in the training set while others end up in the test set, this could result in the classifier being tuned to pick up these time-related similarities among samples instead of (or in addition to) any actually related to workload level. The consequence would be that the true mental state separability could be significantly overestimated.

An alternative approach that mitigates this issue in experiments with this trial structure and associated autocorrelation of samples is to perform block-wise (or trial-wise) cross-validation. In each step of block-wise CV, the trials are first randomly divided into a number of subsets *b*. The samples derived from the trials in one subset are held back for testing, while the samples from the remaining trials are used to train the classifier. This is repeated *b* times until all trials have appeared in the test set exactly once. The overall classifier performance is estimated as the average of the *b* resulting accuracies from each step. This partitioning strategy ensures that all samples from a single trial always remain together in either the training or test set, and, thus, temporal correlations will not influence the results as described above for k-fold CV. k-fold and block-wise cross-validation, as performed on datasets where multiple samples are extracted from a single class’s trials, are illustrated in Figure 2a and Figure 2b, respectively.

While this issue has been acknowledged in some BCI papers [5,6,7] and it has been suggested that block-wise cross-validation should be the preferred approach [5,8], the effects of using k-fold CV in such scenarios has not been investigated, and it is not clear how significantly (and under what circumstances) it can overestimate true mental state separability. Furthermore, it is not clear if block-wise CV will actually accurately estimate class separability. Unfortunately, despite the potential issues, it is still very common to see k-fold CV used in pBCI studies with longer experimental trial structures that allow for block-level temporal correlations to be problematic, so it is important that the effects are understood in order to help inform the best methodology for researchers to use in future studies and to help with the interpretation of the results of past studies that have used this approach.

This study aims to address this knowledge gap. In a preliminary comparison of k-fold and block-wise cross-validation on the classification of EEG data containing block-level temporal correlations, we used two existing publicly available databases: specifically, the SEED [9,10] and DEAP [11] databases. These databases were chosen because they had relatively long trial durations (approximately 1–3 min) for each mental state of interest and allowed us to extract multiple samples/epochs per trial resulting in within-class temporal correlations. Secondly, in order to ascertain the effects related to (1) the “true” task separability, (2) the number of epochs taken from a single trial, and (3) the type of feature/classifier used, we designed an original experiment and collected a new dataset. In this paper, we first present the methodology and results of the preliminary study on the SEED and DEAP databases, and then the methodology and results used in our original study.

## 2. Study 1: Preliminary Analysis on Existing Databases

### 2.1. Materials and Methods

#### 2.1.1. SEED and DEAP Datasets

The SEED dataset [9,10] is a popular EEG dataset for emotion recognition, consisting of fifteen subjects, each participating in three separate sessions. During each session, EEG was recorded while participants observed fifteen movie clips, each chosen to elicit one of three emotional categories: positive, neutral, and negative (five clips per category). The duration of each clip ranged from 3 to 4 min. Data collection was performed via an ESI NeuroScan System using a 62-channel EEG cap with the international 10–20 placement system at a sampling rate of 1000 Hz [9].

For the present study, only the first session from each subject was analyzed, and all trials were truncated to the length of the shortest trial, which was 185 s. Only four of the five trials for each emotion were used to allow for balanced trial-wise randomization. Differential entropy (DE) features were calculated, since they have been shown to be effective for classifying emotional states [10]. Samples for classification were obtained by extracting 185 1-s non-overlapping epochs from each trial, and, for each one, calculating DE in the delta (1–4 Hz), theta (4–8 Hz), alpha (8–12 Hz), beta (12–30 Hz), and gamma (30–50 Hz) frequency bands, for each of the 62 electrodes. This resulted in a 310-dimensional feature vector per sample. The minimum-redundancy maximal-relevance (mRMR) algorithm [12] was used during classification to reduce the feature set dimensionality to 30. The features of the training and testing sets were then z-score normalized seperately, for each fold.

The DEAP dataset [11] is another popular EEG dataset for emotion classification, containing EEG data from 32 participants as they watched 40 1-min music video clips. Unlike the movie clip stimuli used in the SEED experiment, the music clips shown during the DEAP trials were not specifically selected to elicit a particular emotion; instead, each participant rated the level of arousal, valence, like/dislike, dominance, and familiarity for each trial. Data in this dataset were collected using a 32-channel EEG system with the international 10–20 placement at a sampling rate of 512 Hz.

For the present study, to facilitate binary classification, the trials were sorted in order of ascending valence rating and quantized into three levels: the top 10 trials were labelled as positive valence (+), the bottom 10 trials were labelled as negative valence (−), and the middle ten trials (i.e., trials 15–25) were labelled as neutral valence (0). Samples were calculated in the same way that they were for the SEED data: 60 1-s non-overlapping epochs were extracted from each trial, and differential entropy was calculated for the 5 standard frequency bands for each electrode. With 32 electrodes, this resulted in a 160-dimension feature vector for each sample. For consistency with SEED, the mRMR algorithm [12] was again used within each classification fold to reduce the feature set dimensionality to 30. Similar to SEED, the training and testing features were then separately z-score normalized for each fold.

For both the SEED and DEAP datasets, in the analysis described below, all binary combinations of the three emotional states were considered (i.e., positive vs. negative, positive vs. neutral, and negative vs. neutral).

#### 2.1.2. k-Fold and Block-Wise Cross-Validation with True Class Labels

The first step in this preliminary analysis of SEED and DEAP datasets was to compare the classification accuracies estimated via both k-fold and block-wise cross-validation. For both datasets and each of the binary classification problems, 5 runs of 4-fold cross-validation for SEED and 5 runs of 5-fold cross-validation for DEAP were performed on each individual subject’s data. The average of the runs-by-folds accuracies was calculated as an estimate of classifier performance.

Next, for both datasets and each of the binary classification problems, block-wise cross-validation was performed as follows: 5 runs of 4-block cross-validation for SEED and 5 runs of 5-block cross-validation for DEAP. For SEED, this resulted in a block consisting of three complete trials per class used for training and one complete trial per class for testing, and for DEAP, a block containing eight complete trials per class for training and two complete trials per class for testing. Block creation continues until all trials have been used for testing once. As with k-fold cross-validation, the average of the runs-by-blocks accuracies was calculated as an estimate of classifier performance. Note that the number of training and test samples for each step was the same between the k-fold and block-wise CV approaches.

For all scenarios, three different classifiers were considered: linear discriminant analysis (LDA), linear support vector machines (SVM), and k-nearest neighbour (KNN).

#### 2.1.3. k-Fold and Block-Wise Cross-Validation with Randomized Class Labels

While useful, the results of the analysis above only tell us whether the accuracies estimated by the two different cross-validation methods are different from one another, but not whether the k-fold cross-validation is actually over-estimating the accuracy due to the trial structure of the experiment. To further investigate the effect of the temporal correlation among samples extracted from the same trial on the estimation of classification accuracy, we repeated the k-fold and block-wise cross-validation, but this time, we randomly shuffled the class labels first. The class labels were shuffled in two ways:**Trial-level randomization.** In this case, half of the trials from Class 1 were randomly selected, and all samples extracted from those trials were re-labelled as Class 2, after which the same was carried out for Class 2. This completely masked any true differences between the classes, while leaving the temporal correlations related to the trial structure intact;**Sample-level randomization.** In this case, half of the samples from Class 1 were randomly selected and re-labelled as Class 2, and then the same was carried out for Class 2. This again completely mask edany true differences between the classes, but it also eliminated the temporal correlations related to the trial structure.

By combining the two CV techniques and two label randomization techniques, three additional tests of interest were included in the analysis: k-fold CV with sample-level randomization, k-fold CV with trial-level randomization, and block-wise CV with trial-level randomization. Through investigation of these hybrid k-fold and block-wise CV techniques with class label randomization tests, the potential effect of the trial structure on the classifier performance can be seen in the absence of any true class differences.

Again in all scenarios, three different classifiers were considered: linear discriminant analysis (LDA), linear support vector machines (SVM), and k-nearest neighbour (KNN).

### 2.2. Results

Table 1 shows all the classification accuracies (i.e., k-fold CV with true labels, block-wise CV with true labels, k-fold CV with trial-randomized labels, block-wise CV with trial-randomized labels, and block-wise CV with sample-randomized labels), averaged across 32 participants for DEAP and 15 for SEED, for each of the three classifiers (LDA, SVM, KNN) and binary classifications.

Figure 3 depicts a comparison of the classification accuracies for the k-fold and block-wise CVs. For both the SEED and DEAP datasets, the k-fold CV was found via paired *t*-tests to be significantly greater than the block-wise CV for all cases.

Figure 4 compares the random-labelled CV accuracies for the trial-randomized with k-fold, trial-randomized with block, and sample-randomized with k-fold CVs. Significance for these tests was determined via one-sample *t*-tests between the CV accuracies and chance (50%).

For both the DEAP and SEED datasets, trial-randomization with k-fold CV was found to be significantly greater than chance for all scenarios. With any real class differences masked due to the class label randomization, it stands to reason that the classifier had been biased due to “high class-specific temporal correlations” among samples in the training and test sets resulting from the random partitioning of samples.

When using trial-randomization with blocked CV, most scenarios were not found to be significantly different from chance except for three outliers. The positive–neutral classification with the SVM classifier for the DEAP data, and the negative–neutral classification with the SVM classifier and positive–neutral classification with the LDA classifier for the SEED data were all found to be significantly less than chance.

## 3. Discussion

This initial study provides the ground-work evidence of a latent problem within pBCI experiments that use longer duration trials in which many samples are extracted. As previously discussed, k-fold CV is a k-step process in which the samples are randomly divided into *k* partitions (maintaining class balance within each partition), and, at each step, one of the partitions is retained as the test set while the remaining *k*−1 partitions are used to train the classifier. The overall classifier performance is estimated as the average of the resulting *k* accuracies from each step. For both the SEED and DEAP data, relatively long trials of the different mental states of interest were collected, and we have extracted many different samples (60 for DEAP, 185 for SEED) from each trial. Thus, the random partitioning of the data in k-fold cross-validation will result in the test set containing samples that are highly temporally correlated with samples in the training set. The concern is that this will bias the classifier and yield inflated classification accuracies that do not necessarily reflect the true separability of the mental states.

This concern was shown to be real, as can be seen when comparing the accuracies of the k-fold and blocked CV methods in Figure 3. Across all classification categories for the SVM classifier, with true class labels, the blocked CV accuracy was found to be significantly lower than its k-fold counterpart. Providing additional, and perhaps even more compelling, evidence of the temporal correlation effects were the randomized label tests, as seen in Figure 4. Whenever k-fold CV was used after trial-level class label randomization, results were found to be significantly greater than chance. With the SEED database, the k-fold CV results for the true and trial-randomized labels were all within approximately 5% of each other, and for the negative/neutral classification, the classification accuracy for the randomized labels was actually 1.2% higher than for the true labels (rand: 92.1%; true: 90.9%). With all real class differences masked due to class label randomization, these randomized test results indicate that k-fold CV results can be significantly inflated entirely due to the trial structure and associated underlying autocorrelation of the samples within trials. This casts doubt on the reliability of classification results obtained via k-fold cross-validation under similar experimental conditions.

## 4. Study 2: Original Experiment

### 4.1. Objectives

While Study 1 was able to show the presence of time-correlation effects on the cross-validation processes, the pre-existing datasets used were insufficient to be able to provide answers as to how much of an effect trial length, among other factors, has on these cross-validation techniques. As such, an original study was conducted to better answer these questions. By using multiple trial lengths within the same experiment, this study aimed to determine how trial length and class separability influence the outcomes of cross-validation techniques, how susceptible common classifiers and features sets are to time correlations in cross-validation, and whether current cross-validation techniques are over- or under-estimating a potential “ground-truth” accuracy.

### 4.2. Materials and Methods

#### 4.2.1. Participants

In all, 12 healthy participants were recruited for this study (aged 24.75 ± 1.64 years; 11 male; 11 right-handed). Inclusion criteria required participants to be between 18 and 65 years old, have normal or corrected-to-normal vision and hearing, have no history of neurological disease, disorder, or injury, and have no cognitive impairment. Participants were asked not to exercise, smoke, or consume caffeine, alcohol, or other drugs within four hours of starting the experimental session. All participants provided written informed consent before participating. This study was approved by the Interdisciplinary Committee on Ethics in Human Research (ICEHR) at Memorial University of Newfoundland, NL, Canada.

#### 4.2.2. Instrumentation

EEG signals were acquired via a 64-channel actiCHamp system ( Brain Products GmbH, Gilching, Germany) at a sampling rate of 500 Hz. Electrodes were placed according to the international 10–10 system. The reference and ground electrodes were set as FCz and FPz, respectively. Electrode impedances were initially lowered to ≤10 kΩ and were then checked periodically during the session and reduced as necessary.

#### 4.2.3. Experimental Protocol

Following set-up of the EEG system, participants were asked to complete two one-minute baseline trials—one with eyes open and one with eyes closed. During the rest of the session, participants completed three different types of trials: reading, listening, and rest. In order to investigate the effect of increasing the number of samples extracted from the same trial on the performance of the different cross-validation methods, 3 different trial lengths $T_{n}$ (for *n* = 5, 15, and 60 s) were considered. The session was divided into three main sections, one for each of the trial lengths. There are six possible permutations of the three lengths, so each participant was pre-assigned a unique order in which they would complete the three sections (each permutation was completed by two participants).

For each section (i.e., trial length), a total of six minutes of data were recorded for each of the three tasks. These six minutes were divided into trials of the appropriate length for that section (i.e., for the $T_{5}$, $T_{15}$, and $T_{60}$ sections, there were 72, 24, and 6 trials per task, respectively). Each section was divided into three blocks (two minutes of recording per task, per block). Participants were allowed to rest as needed between blocks. The order of presentation of the task trials was random in each section.

Trials began with an image and text cue, indicating the upcoming trial type and prompting the participant to press space when ready to start. Upon pressing space, a one-second blank transition screen appeared, followed by the appropriate task stimulus (described below) for $T_{n}+1$ s. A one-second buffer (blank screen) was presented at the end, after which the next trial’s prompt appeared. This protocol is summarized in Figure 5.

For the reading tasks, a passage (in English) was presented on the screen during the stimulus period. Participants were instructed to read at a comfortable pace in their head (not out loud) until the trial ended. The text passages, which were gathered from online free-to-use paragraph generators, were carefully selected to take slightly longer than the average adult reading speed for the given trial length; however, if a participant did finish before the end of the trial, they were asked to start reading again from the beginning of the passage. Auditory stimuli for the listening trials were generated via a paid online text-to-speech bot (from the same text passages used for the reading trials) so that all trials would have a consistent pacing, volume, and pronunciation. During these trials, the passages were played through computer speakers placed on the desk in front of the participant. For rest trials, a cross was presented on a blank screen for the participants to focus on (this cross was presented for the listening trials as well).

#### 4.2.4. Data Analysis

##### Pre-Processing

All pre-processing was performed on the aggregate of each participant’s data. Pre-processing was completed in Matlab using the package EEGLab [13]. The following steps were used:Downsampling from 500 Hz to 250 Hz;Bandpass filtering from 0.5 Hz to 55 Hz;Artifact Subspace Reconstruction (ASR), using default settings, to remove channels that were poorly correlated with adjacent channels and to remove non-stationary high-amplitude bursts;Interpolating channels removed from ASR;Re-referencing data to the common average.

##### Feature Calculation

After pre-processing, the EEG data from each trial (of all trial lengths) were then segmented into five-second non-overlapping epochs. Next, the following features were calculated over each epoch: band power in the delta (1–4 Hz), theta (4–8 Hz), alpha (8–12 Hz), beta (12–30 Hz), and gamma bands (30–50 Hz), spectral entropy in the delta band, and root-mean-square (RMS) and variance across the entire spectrum (0.5–55 Hz), as they have been shown previously to be effective for reading tasks [14,15]. All electrodes (excluding FPz as ground) were used for feature calculation. All features were z-score normalized based on the training set during each fold/block of cross-validation.

Note that samples were extracted from 5-s non-overlapping epochs because 5 s was the shortest trial length recorded. The $T_{5}$ trials were used to represent the “ground truth” class separability because, for these trials, only 1 sample was extracted per trial and because the task order was random; this means that, for the $T_{5}$ trials, the samples used for classification were completely randomized between classes across time (and thus contain no class-specific temporal correlations). The $T_{15}$- (with 3 samples extracted per trial) and $T_{60}$ (with 12 samples extracted per trial)-length trials are meant to represent datasets with relatively low and relatively high (respectively) temporal correlations among samples within a class.

##### Classification: k-Fold and Block-Wise Cross-Validation

For each classification scenario considered, the following analyses were performed:

**k-fold cross-validation:** Samples were randomly divided into six (i.e., k = 6) equal subsets (balanced between classes). One of the subsets was retained for testing, while the remaining five subsets were used to train the classifier. Classification accuracy was then calculated. This was repeated six times, until all subsets (and thus, all individual samples) were used for testing the classifier exactly once. This whole process, starting with the random division of the samples into six subsets, was repeated five times. The overall estimate of classification accuracy was calculated as the average of the 30 resulting classification accuracies.

This six-fold CV process was performed with (1) true class labels, (2) trial-randomized class labels (see Section 2.1.3)), and (3) sample-randomized class labels (see Section 2.1.3);

**Block-wise cross-validation:** Trials were randomly divided into six equal subsets (balanced between classes). All samples from the trials in one of the subsets were retained for testing, while the samples from the trials in the remaining five subsets were used for training the classifier. Classification accuracy was then calculated. This was repeated until all subsets (and thus, all individual samples) were used for testing the classifier exactly once. This whole process, starting with the random division of the trials into six subsets, was repeated five times. The overall estimate of classification accuracy was calculated as the average of the 30 resulting classification accuracies. Unlike in the six-fold CV described above, in this approach, all samples extracted from a given trial always remain together in either the training or the test set.

This process was performed with (1) true class labels and (2) trial-randomized class labels (see Section 2.1.3).

##### Classification: Factors Considered

To move towards the objectives of Study 2, we investigated how the following factors influence the effect that temporal correlations have on the classification results from the k-fold and block-wise CV approaches:**Class separability:** Two different task pairs were considered: (1) read vs. rest, which represented the high-separability case, and (2) listen vs. rest, which represented the low separability case. These task pairs were chosen based on the results of pilot data; in preliminary analysis, the read vs. rest and listen vs. rest pairs were found to be separable, with approximately 96% and 68% accuracy, respectively, as determined using bandpower features and an SVM classifier. Note that read vs. listen was initially considered as well, but, because it was determined to have approximately the same separability as read vs. rest and, thus, provided no additional insights related to our research questions, it is not reported here;**Amount of class-specific temporal correlation:** Classification of the data from each trial length was considered separately, so that the results from each could then be compared. Because six minutes of data were collected (per task) for each trial length, once divided into five-second epochs, the number of samples used for classification was identical for the three different trial lengths. The only difference was the amount of temporal correlation that existed within the data for the 3 conditions; the 15 s length (with 3 samples per trial) represented relatively low temporal correlation, while the 60 s length (with 12 samples per trial) represented relatively high temporal correlation;**Feature types:** All calculated feature categories (i.e., band power, spectral entropy, RMS, variance) were treated separately and were not combined for classification. For all classification problems considered, feature set dimensionality was reduced to 30 via the mRMR algorithm [12] (this was performed using just the training set at the appropriate time within the cross-validation analyses);**Classifier:** The same as in the first study, three different classifiers were investigated: LDA, SVM, and KNN. Hyperparameters were optimized for each classification using Matlab’s default automatic hyperparameter optimizer.

To summarize, we investigated the combination of 2 binary classification problems (the low and high separability cases), 4 feature sets (bandpower, spectral entropy, RMS, variance), and 3 classifiers (SVM, LDA, KNN), for a total of 24 classification scenarios. When considering the individual cross-validation tests, 5 were performed for the 15 and 60 s trial lengths (2 for true labels, 3 for randomized labels), and 2 were performed for the 5-s trial length (one for each the true and randomized labels, because k-fold and blockwise CV are identical in this case, as are sample-wise and trial-wise label randomization). This resulted in 288 individual cross-validation tests per participant.

#### 4.2.5. Statistical Analysis

To better understand if and how factors such as amount of temporal correlation and true class separability influence the effect of the temporal correlation on the k-fold and block-wise cross-validation approaches, the following statistical tests were performed for every combination of class separability (low and high), amount of class-specific temporal correlation (3 samples per trial and 12 samples per trial), classifier, and feature set:

Classification with true labels: Accuracies for the k-fold and block-wise CV approaches were compared to the ground truth accuracy (as determined by the 5-second trials) via individual paired *t*-tests;

Randomized labels: Accuracies for the k-fold CV with trial-randomization and block-wise CV with trial-randomization were each compared to chance (50% for binary classification) via one-sample *t*-tests. Note that for the randomized label scenarios, the “ground truth” class separability is chance.

### 4.3. Results

Table 2 shows the most pertinent classification results (i.e., ground truth, k-fold CV with true labels, block-wise CV with true labels, k-fold CV with trial-randomized labels, and block-wise CV with trial-randomized labels) for all combinations of class separability, trial length, classifier, and feature set. For the “true-label” scenarios, over-estimations and under-estimations from the k-fold and block-wise CVs compared to the ground-truth are represented in blue and red, respectively. For the trial-randomized k-fold CV and trial-randomized block-wise CV tests, blue and red are used to indicate over-estimation and under-estimation compared to chance (i.e., 50%). Statistically significant results, as determined by the statistical tests described in Section 4.2.5, are denoted by bold font. Note that, for the k-fold CV with sample-randomized labels analysis, as expected, all scenarios were not significantly different from chance (average accuracies ranged from 48.4–52.0%, so they are not shown in the table).

To aid in visualization of the trends in the different cross-validation analysis for the factors of class separability and amount of temporal correlation, Figure 6 represents all classification results for the SVM classifier and bandpower feature set (general trends across these factors were similar for all classifier/feature set combinations).

### 4.4. Discussion

This study explored the limitations of conventional cross-validation techniques on single-subject EEG signal classification in scenarios where there was time-related correlation among samples within a class. Such scenarios are very common in offline passive BCI studies where, typically, relatively long trials of different mental states (e.g., low and high workload) are recorded, and multiple short epochs are extracted from these trials and serve as samples for classification. In this scenario, the use of k-fold cross-validation is problematic because the randomized division of samples into the k subsets means that, in any given “fold”, some samples from a single trial end up in the training set while others end up in the test set. Additionally, because samples from a single trial will be highly correlated due to their proximity in time, this can potentially influence the classification accuracy and cause over-estimation of the true class separability. An alternative to k-fold CV is “block-wise” or “trial-wise” cross-validation, where trials are randomly divided into subsets, and short epochs are extracted from these trials to serve as samples for classification. In this case, all samples from a single trial always stay together in either the training or test set, and so the problem inherent in k-fold cross-validation should be eliminated. To the best of our knowledge, however, the actual effects of using k-fold CV in such scenarios had not been investigated, and it was not clear how significantly (and under what circumstances) it can overestimate true mental state separability. Furthermore, it was not clear if block-wise CV actually accurately estimates class separability. The motivation behind this work was to investigate the extent to which standard cross-validation techniques may misrepresent “true” class separability of EEG data, whether that may be over- or under- estimation, in scenarios where time-related correlations exist among samples within a class. The tests used in this study could be broadly applied to any EEG dataset using single-subject classification and a similar trial structure, to gauge the potential impact on the results.

For the case of high true class separability, k-fold CV produced results very close to the “ground truth” accuracies (as determined by the 5-s trials) which were between approximately 88% and 95%, depending on the specific classifier and feature set used. This was true for both the “low class-specific temporal correlations” and “high class-specific temporal correlations” conditions. There were only a few scenarios where the k-fold CV accuracy was significantly different from the ground-truth according to a paired *t*-test (and, in these cases, the k-fold CV result did not overestimate, but rather was actually *less* than the ground truth by about 2%). In the high true class separability case, block-wise CV did not perform quite as well as k-fold CV; while many results were fairly close, there were more scenarios where the results were significantly less than the “ground truth” accuracies. This was more pronounced for the case of “high class-specific temporal correlations,” where the blockwise CV result was, on average, 5% less than the ground-truth.

For the case of low true class separability, the amount of class-specific temporal correlation strongly affected the outcomes of the two CV methods. When there are low amounts of class-specific temporal correlation, both k-fold and block-wise CV provided results similar to the “ground-truth.” There were only three scenarios which were found to be significantly different from the ground-truth according to paired *t*-tests, two of which were over-estimations by k-fold CV (by about 4%) and one of which was under-estimation by block-wise CV (by about 5%). On average, k-fold and block-wise CV over- and under- estimated the ground-truth by 2.5% and 0.6%, respectively.

Lastly, for the case of low true class separability and high amounts of class-specific temporal correlation, the majority of both k-fold and block-wise CV methods were significantly different from the ground-truth. For k-fold CV, 11 of 12 scenarios were found to significantly overestimate the ground-truth and by a substantially greater degree (significant overestimations ranged from 7.86% to 12.09%) than in the previous case of low amounts of class-specific temporal correlation. For block-wise CV, 8 of 12 scenarios were found to significantly underestimate the ground truth, also by a substantially greater degree than the previous case (significant underestimations ranged from −6.05% to −11.11%).

Broadly speaking, the case of low class separability with low amounts of class specific temporal correlation (where we only used three samples per trial) performed adequately in comparison to the ground-truth; however, when high amounts of class-specific temporal correlation are present (for this study, 12 samples per trial), one can begin to see how k-fold CV becomes unreliable as the temporal correlation begins inflating the accuracies. To further highlight how the amount of class-specific temporal correlation can inflate accuracies, the randomized label tests from Section 2.1.3 were also used—of specific note are the trial-randomized k-fold (t.r. k-f) tests. This combination effectively negates any true class differences while retaining the amount of class-specific temporal correlation, and, in Table 2, all 24 scenarios of t.r. k-f were found to be significantly greater than chance. For the case of low true class separability, the t.r. k-f CV accuracies were often within a few percentage points of the true labeled k-fold CV accuracies, thus casting significant doubt on the veracity of those accuracy figures.

As mentioned in Section 4.3, epoch-level label randomization with k-fold CV tests were also performed to verify that, when no class differences and no time correlation were present, the outcomes of classifications were consistently chance. Trial-level label randomization with block-wise CV tests were performed to verify if block-wise CV was able to negate the time correlations present when using k-fold CV. This did generally result in accuracies much closer to chance, but the accuracies did not always average to exactly chance, rather showing some over- and under-shooting. It could be that block-wise CV introduces additional shortcomings into the cross-validation process that are not fully understood.

For the true labelled tests, the magnitude of over- and underestimation varied by feature. The bandpower and spectral entropy features consistently overestimated (with k-fold CV) and underestimated (with block-wise CV) the ground-truth accuracy by the greatest margins. On the other hand, RMS features consistently underestimated the ground-truth by a smaller margin when using block-wise CV, as compared to the other features, such that it was never found to be significantly different from the ground-truth in any of low class separability scenarios, regardless of the amount of class-specific temporal correlation. The choice of classifier also appeared to have a slight impact on the extent to which k-fold CV overestimated and block-wise CV underestimated the ground-truth accuracy, as can be seen in the feature average rows of Table 2. In the low class separability/long trial length case, where the differences from ground-truth were the greatest across the board, the overestimatation via k-fold CV and the underestimatation via block-wise CV were both larger on average for the KNN classifier than for either SVM or LDA. Given our previous explanation of the issue arising due to time-related correlations among samples from a single trial, it is not surprising that a classifier that predicts the class of a test sample based on the class membership of the nearest training samples would not be reliable.

With the evidence collected throughout this original study, it is the recommendation of this paper that, when reporting the results of analysis based on single-subject classification for pBCI applications that use a longer trial duration, authors should provide both the k-fold and block-wise cross-validation accuracies, as well as the accuracies for trial-level label randomization with k-fold cross-validation. Together, these metrics provide stronger evidence regarding the efficacy of new neural indicators presented to the field, as they indicate the influence of the possible compounding factor of time correlation between samples due to trial duration and experimental protocol. Nevertheless, it is worth nothing that, in passive BCIs, the block-wise CV scenario more accurately reflects the training/testing conditions for an online BCI, so even if it does underestimate the "true" class separability, it might represent what is practically achievable. Authors should also carefully consider trial duration during study design and attempt to reduce the duration of individual trials as much as possible, as this has been seen to have a positive effect on reducing time correlation effects.

This original experiment allowed us to compare the results obtained from k-fold and block-wise cross-validation to a “ground truth” class separability and clearly showed the impact of within-class temporal correlation on the reliability of these results. We were also able to investigate several different pertinent factors that could further affect the outcomes of k-fold and block-wise CV under these scenarios—in particular, the effect of the amount of temporal correlation and the degree of true class separability. However, due to practical limitations on the duration of the experimental session, we were only able to use three different trial durations, representing three different “amounts” of temporal correlation, and also could only investigate three different levels of class separability (e.g., not separable/random, low separability, high separability). It would have been ideal to use additional trial durations and additional degrees of true separability to give a more precise picture of the conditions under which k-fold cross-validation results in a statistically significant divergence from the GT. It may be worth further investigating these factors in the future.

While we have focused on the common passive BCI experimental paradigm where multiple EEG samples from a given mental state are derived from a single long trial, this issue would also exist for paradigms where a single sample is taken per trial, but where the trials are not completely randomized across states/classes. Additionally, the results of this study are likely relevant to other commonly used physiological time-series signals, such as ECG, EMG, and fMRI, to name a few examples. Further investigation with these signal types may help to broaden the understanding of this issue.

## 5. Conclusions

We have evaluated two popular pBCI datasets (SEED and DEAP) using multiple cross-validation approaches and tests. For both of these datasets, significant time-correlation effects were found, substantially more so for the SEED dataset which used longer trial durations (SEED: 185s, DEAP: 60s). We have also conducted an original study in which 62-channel EEG signals were recorded from 12 participants while they read and listened to selected text and audio prompts for three different trial durations (5 s, 15 s, and 60 s). After training three common classifiers (SVM, LDA, KNN) on four proposed features (bandpower, spectral entropy, RMS, variance), for each trial duration, the various cross-validation procedures were evaluated. The experimental results showed that, in cases of high class separability, k-fold cross-validation produced a reasonably close estimate of the proposed ground-truth, while block-wise cross-validation produced a significant underestimation of the ground-truth that increased in significance as trial duration increased. In cases of low class separability and longer trial durations, it was evident that the time correlations can begin to supersede the inherent class separability and cause an overestimation in classification accuracy when using standard k-fold cross-validation, which could lead to faulty conclusions to be drawn about the data. The results also showed that block-wise cross-validation may not be a perfect k-fold substitute, as, when used in the previous conditions, block-wise CV was found to underestimate the proposed ground-truth.

## Figures and Tables

**Figure 1 sensors-23-06077-f001:**
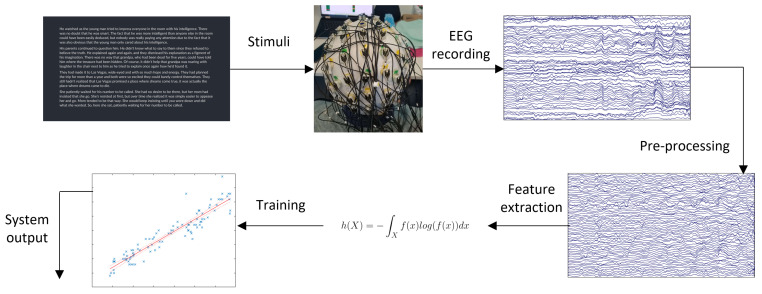
BCI system overview.

**Figure 2 sensors-23-06077-f002:**
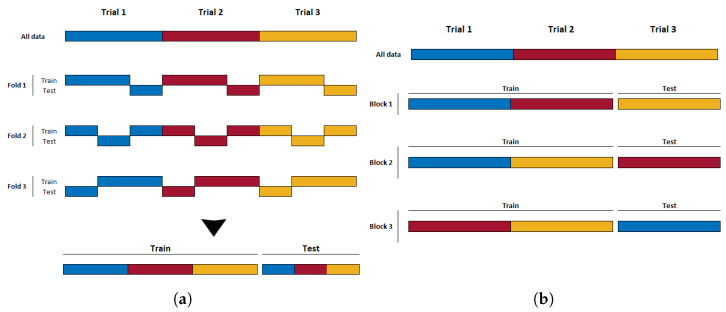
(**a**) An example of k-fold CV; epochs from a single trial end up being mixed into both the training and testing sets. (**b**) An example of block-wise CV; by not breaking up the trial structure, epochs from a given trial remain exclusively in either the training or testing set.

**Figure 3 sensors-23-06077-f003:**
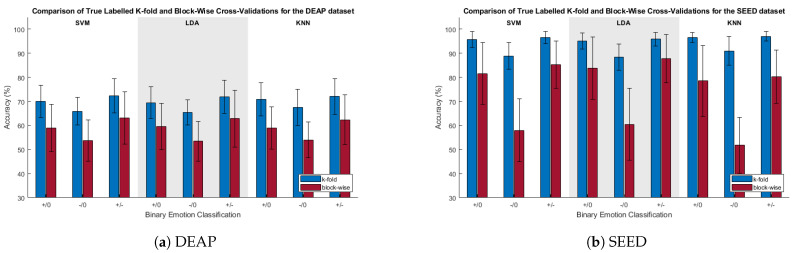
Comparison of true-labeled classification accuracies when using k-fold and blocked CV. +/0, −/0, and +/− are the positive/neutral, negative/neutral, and positive/negative classifications, respectively.

**Figure 4 sensors-23-06077-f004:**
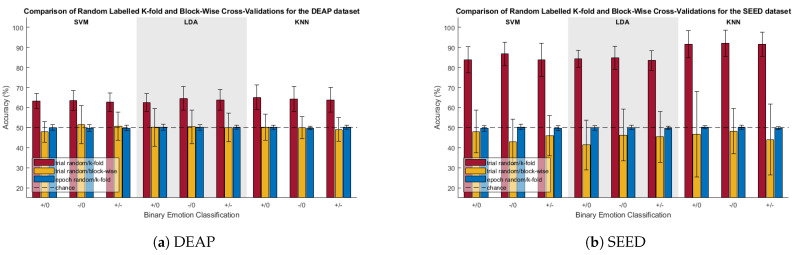
Comparison of randomized-labeled classification accuracies of k-fold and blocked CV. +/0, −/0, and +/− are the positive/neutral, negative/neutral, and positive/negative classifications, respectively.

**Figure 5 sensors-23-06077-f005:**
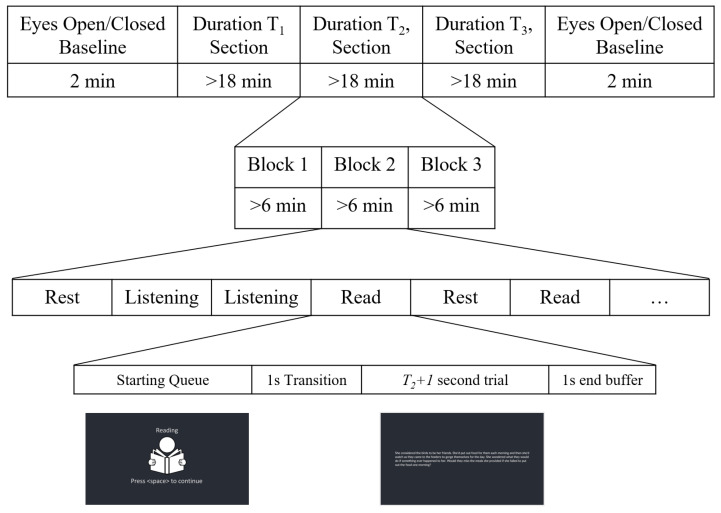
Overview of the experimental protocol for recording multiple trial lengths.

**Figure 6 sensors-23-06077-f006:**
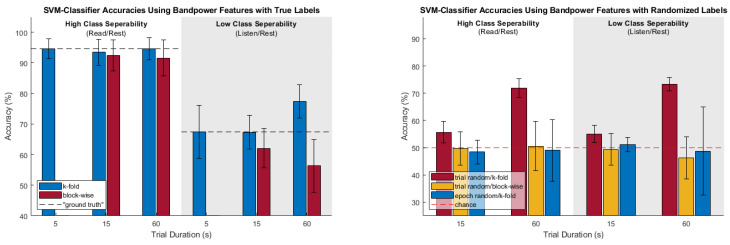
SVM bandpower accuracies for true and random labelled cross validations.

**Table 1 sensors-23-06077-t001:** Classification accuracies and statistical significance for the primary tests on DEAP and SEED. Columns denoted by ${}^{1}$ were compared together via a paired *t*-test, and columns denoted by ${}^{2}$ were compared to chance (50%) via one-sample *t*-tests. Bolded *p*-values indicate statistical significance. Under randomized labels, t.r. k-f, t.r. bl., and s.r. k-f are short for trial-randomized k-fold, trial-randomized block-wise, and sample-randomized k-fold CV, respectively.

			True Labels			Random Labels					
Dataset	Classifier	Classification	k-Fold ${}^{1}$	Block ${}^{1}$	${}^{1}$	t.r. k-f ${}^{2}$		t.r. bl. ${}^{2}$		s.r. k-f ${}^{2}$	
DEAP	SVM	Pos–Neu	70.00	58.91	***p* < 0.001**	63.28	***p* < 0.001**	47.91	***p* = 0.027**	50.05	*p* = 0.830
		Neg–Neu	65.90	53.75	***p* < 0.001**	63.44	***p* < 0.001**	51.43	*p* = 0.400	49.82	*p* = 0.533
		Pos–Neg	72.29	63.13	***p* < 0.001**	62.70	***p* < 0.001**	50.82	*p* = 0.512	49.84	*p* = 0.478
	LDA	Pos–Neu	69.44	59.63	***p* < 0.001**	62.49	***p* < 0.001**	50.16	*p* = 0.922	50.19	*p* = 0.487
		Neg–Neu	65.37	53.42	***p* < 0.001**	64.57	***p* < 0.001**	50.39	*p* = 0.793	50.14	*p* = 0.553
		Pos–Neg	71.83	62.81	***p* < 0.001**	63.86	***p* < 0.001**	50.06	*p* = 0.964	50.16	*p* = 0.378
	KNN	Pos–Neu	70.83	58.89	***p* < 0.001**	65.07	***p* < 0.001**	50.26	*p* = 0.823	50.17	*p* = 0.391
		Neg–Neu	67.45	54.01	***p* < 0.001**	64.30	***p* < 0.001**	50.03	*p* = 0.972	49.84	*p* = 0.371
		Pos–Neg	72.02	62.31	***p* < 0.001**	63.80	***p* < 0.001**	49.05	*p* = 0.370	50.21	*p* = 0.212
SEED	SVM	Pos–Neu	95.70	81.57	***p* < 0.001**	83.86	***p* < 0.001**	48.03	*p* = 0.484	49.63	*p* = 0.317
		Neg–Neu	88.89	57.95	***p* < 0.001**	86.75	***p* < 0.001**	42.88	***p* = 0.028**	49.63	*p* = 0.480
		Pos–Neg	96.54	85.22	***p* < 0.001**	83.83	***p* < 0.001**	45.97	*p* = 0.143	49.71	*p* = 0.351
	LDA	Pos–Neu	95.10	83.80	***p* < 0.001**	84.21	***p* < 0.001**	41.35	***p* = 0.018**	49.88	*p* = 0.699
		Neg–Neu	88.42	60.48	***p* < 0.001**	84.79	***p* < 0.001**	46.25	*p* = 0.278	50.06	*p* = 0.840
		Pos–Neg	95.85	87.86	***p* = 0.001**	83.44	***p* < 0.001**	45.44	*p* = 0.185	49.75	*p* = 0.288
	KNN	Pos–Neu	96.44	78.51	***p* < 0.001**	91.52	***p* < 0.001**	46.70	*p* = 0.560	50.21	*p* = 0.358
		Neg–Neu	90.94	51.73	***p* < 0.001**	92.14	***p* < 0.001**	48.17	*p* = 0.541	50.11	*p* = 0.723
		Pos–Neg	97.04	80.24	***p* < 0.001**	91.53	***p* < 0.001**	44.08	*p* = 0.217	49.83	*p* = 0.386

**Table 2 sensors-23-06077-t002:** Classification accuracies for all classifiers and feature sets. Blue/red intensities denote the overestimation/underestimation of the CV techniques compared to the ground truth (G.T.) for k-fold and block columns, and compared to chance (50%) for trial-random k-fold (t.r. k-f) and trial-random block (t.r. bl.) columns. Heat map intensities are calculated separately for the ranges in ${}^{1}$, ${}^{2}$, and ${}^{3}$. Bolded values were found to be significantly different, as per Section 4.2.5.

			High Seperability					Low Seperability				
Classifier	Trial Duration	**Feature**	G.T.	k-Fold ${}^{1}$	Block ${}^{1}$	t.r. k-f ${}^{2}$	t.r. bl. ${}^{3}$	G.T.	k-Fold ${}^{1}$	Block ${}^{1}$	t.r. k-f ${}^{2}$	t.r. bl. ${}^{3}$
SVM	15 s	Bandpower	94.58	93.43	92.41	**55.67**	49.73	67.41	67.34	**62.07**	**55.06**	49.32
		Spectral Ent.	95.02	93.68	**92.29**	**57.14**	49.21	65.71	67.36	64.62	**55.08**	51.72
		RMS	91.53	90.09	**89.12**	**55.38**	49.54	62.14	66.64	63.11	53.41	46.96
		Variance	91.83	90.29	**89.11**	52.92	50.07	62.18	65.59	63.84	52.85	48.84
		15 s avg	93.24	91.87	90.73	55.28	49.64	64.36	66.73	63.41	54.10	49.21
	60 s	Bandpower	94.58	94.58	**91.57**	**71.82**	50.61	67.41	**77.37**	**56.30**	**73.33**	46.22
		Spectral Ent.	95.02	94.43	**91.17**	**70.84**	45.53	65.71	**77.80**	**55.16**	**71.84**	58.40
		RMS	91.53	90.47	**86.53**	**62.52**	**42.67**	62.14	**72.05**	59.51	**64.47**	48.56
		Variance	91.83	90.58	**86.62**	**62.40**	**41.74**	62.18	**71.16**	58.50	**60.37**	53.85
		60 s avg	93.24	92.52	88.97	66.90	45.14	64.36	74.59	57.37	67.50	51.76
LDA	15 s	Bandpower	95.05	93.23	93.14	**55.96**	51.78	66.72	67.55	64.54	**55.87**	47.82
		Spectral Ent.	95.24	93.32	**92.97**	**54.98**	50.34	66.18	68.26	65.52	54.46	52.62
		RMS	92.55	**90.31**	**89.55**	53.54	47.60	63.18	66.46	63.76	**53.15**	**46.82**
		Variance	91.67	**89.44**	**88.32**	**53.44**	48.37	64.10	67.15	64.36	53.41	51.68
		15 s avg	93.63	91.58	91.00	54.48	49.52	65.05	67.36	64.55	54.22	49.73
	60 s	Bandpower	95.05	94.05	**91.00**	**73.58**	50.39	66.72	**78.08**	**56.97**	**73.45**	48.43
		Spectral Ent.	95.24	93.92	**90.60**	**72.50**	**40.96**	66.18	**77.89**	**57.78**	**72.25**	55.96
		RMS	92.55	91.60	**87.63**	**63.61**	46.83	63.18	**72.63**	61.47	**64.99**	48.62
		Variance	91.67	90.59	85.74	**64.26**	**43.47**	64.10	**71.96**	**57.91**	**65.07**	51.24
		60 s avg	93.63	92.54	88.74	68.49	45.41	65.05	75.14	58.53	68.94	51.06
KNN	15 s	Bandpower	92.85	92.23	91.76	**54.06**	50.39	62.78	65.06	61.50	**54.86**	50.00
		Spectral Ent.	93.09	92.75	91.93	**57.57**	**46.70**	62.67	**66.57**	62.67	**56.08**	49.58
		RMS	88.51	86.22	84.86	**56.04**	49.71	59.19	**63.00**	59.86	**51.81**	47.99
		Variance	87.77	85.82	84.72	**53.74**	49.19	60.87	62.55	59.84	**53.47**	51.94
		15 s avg	90.55	89.26	88.32	55.35	49.00	61.38	64.29	60.97	54.05	49.88
	60 s	Bandpower	92.85	93.65	88.53	**78.67**	**41.70**	62.78	**78.00**	**51.50**	**76.66**	51.27
		Spectral Ent.	93.09	93.84	**88.32**	**75.10**	48.26	62.67	**74.71**	**53.83**	**73.56**	46.25
		RMS	88.51	87.29	**81.34**	**64.70**	45.24	59.19	**67.55**	55.71	**65.25**	48.82
		Variance	87.77	87.57	**80.72**	**64.19**	47.28	60.87	67.08	**54.81**	**62.94**	49.13
		60 s avg	90.55	90.59	84.73	70.67	45.62	61.38	71.83	53.96	69.60	48.87

## Data Availability

The datasets presented in this article are not readily available because release of study data requires approval of the local ethics authority. Requests to access the datasets should be directed to Sarah D. Power, sd.power@mun.ca.

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
