# Peer review of "k-Fold Cross-Validation Can Significantly Over-Estimate True Classification Accuracy in Common EEG-Based Passive BCI Experimental Designs: An Empirical Investigation"

_sensors, 2023, doi:10.3390/s23136077_

Round 1
Reviewer 1 Report
This paper studies the impact of k-fold cross-validation on the performance of the brain-computer interface (BCI). The topic is timely and suitable for publication in Sensors. Some weak points are given below:
Other metrics besides k-fold and block-wise cross-validation should be considered to enhance the manuscript’s contributions.
There lack of some baseline mathematical framework to explain results, and get insights into some outcomes.
The motivations for choosing two data sets are not well-presented
The presentation needs to be seriously updated, many abbreviations in the Abstract are used without definitions, and the first letter of the first word of some sentences is not in capital, etc.
1. What are the strengths and weaknesses of the proposed approach?
2. Apart from the two adopted datasets, do you compare them with other popular datasets? Also, the contributions will be significantly enhanced if the authors compare with other state-of-the-art under the same dataset?
3. The reviewer suggests adding a table to summarize all notations used in the manuscript.
4. The abstract should add more results and discusses their significance
English is readable and understandable but proofread is required.
Reviewer 2 Report
The proposed article focuses on the evaluation of k-fold cross-validation usage for separating different mental states, in passive BCI studies. An experiment is proposed to evaluate how much the degree of correlation between samples in the same class, alter the accuracy of classification of different mental states from EEG signals.
I have no major comments
Minor comments:
1. CV abbreviation in abstract should be explained.
2. Introduce number for the figure in page 3 and revise the caption
3. Capital R for the beginning of the sentence in row 216.
